# Deciphering the functional and structural complexity of the Solar Lake flat mat microbial benthic communities

Rehab Z. Abdallah,[1] Ali H. A. Elbehery,[2] Shimaa F. Ahmed,[1] Amged Ouf,[1] Mohamed N. Malash,[3] Werner Liesack,[4] Rania Siam[1]

**ABSTRACT** The Solar Lake in Taba, Egypt, encompasses one of the few modern-day microbial mats' systems metabolically analogous to Precambrian stromatolites. Solar Lake benthic communities and their adaptation to the Lake's unique limnological cycle have not been described for over two decades. In this study, we revisit the flat mat and describe the summer's shallow water versus exposed microbial community; the latter occurs in response to the seasonal partial receding of water. We employed metagenomic NovaSeq-6000 shotgun sequencing and 16S rRNA, *mcrA,* and *dsrB* quantitative PCR. A total of 292 medium-to-high-quality metagenome-assembled genomes (MAGs) were reconstructed. At the structural level, *Candidatus* Aenigmatarchaeota, Micrarchaeota, and Omnitrophota MAGs were exclusively detected in the shallow-water mats, whereas *Halobacteria* and *Myxococcota* MAGs were specific to the exposed microbial mat. Functionally, genes involved in reactive oxygen species (ROS) detoxification and osmotic pressure were more abundant in the exposed than in the shallow-water microbial mats, whereas genes involved in sulfate reduction/oxidation and nitrogen fixation were ubiquitously detected. Genes involved in the utilization of methylated amines for methane production were predominant when compared with genes associated with alternative methanogenesis pathways. Solar Lake methanogen MAGs belonged to *Methanosarcinia*, *Bathyarchaeia*, *Candidatus* Methanofastidiosales*,* and *Archaeoglobales*. The latter had the genetic capacity for anaerobic methane oxidation. Moreover, *Coleofasciculus chthonoplastes*, previously reported to dominate the winter shallow-water flat mat, had a substantial presence in the summer. These findings reveal the taxonomic and biochemical microbial zonation of the exposed and shallow-water Solar Lake flat mat benthic community and their capacity to ecologically adapt to the summer water recession.

**IMPORTANCE** Fifty-five years ago, the extremophilic "Solar Lake" was discovered on the Red Sea shores, garnering microbiologists' interest worldwide from the 1970s to 1990s. Nevertheless, research on the lake paused at the turn of the millennium. In our study, we revisited the Solar Lake benthic community using a genome-centric approach and described the distinct microbial communities in the exposed versus shallow-water mat unveiling microbial zonation in the benthic communities surrounding the Solar Lake. Our findings highlighted the unique structural and functional adaptations employed by these microbial mat communities. Moreover, we report new methanogens and phototrophs, including an intriguing methanogen from the *Archaeoglobales* family. We describe how the Solar Lake's flat mat microbial community adapts to stressors like oxygen intrusion and drought due to summer water level changes, which provides insights into the genomic strategies of microbial communities to cope with altered and extreme environmental conditions.

**KEYWORDS** Solar Lake, hypersaline, community genomics, methanogenesis, candidate phyla, flat microbial mat

Address correspondence to Rania Siam, rsiam@aucegypt.edu.

The authors declare no conflict of interest.

See the funding table on p. 17.

Hypersaline lakes are water bodies of significant ecological importance due to their classification as "poly-extreme ecosystems." These lakes inhabit harsh environmental conditions, including high salinity levels above 35 g/L, exposure to UV radiation, and low dissolved oxygen (1). Hypersaline lakes harbor distinctive microbial mat ecosystems characterized by high microbial diversity, environmental resilience, and adaptability. These microbial mats have been previously studied in Shark Bay, Australia, Lake Fryxell, the Arctic, Guerrero Negro, Mexico, Kiritimati Atoll, Pacific Ocean (2–6), and the Solar Lake, Egypt (7–11).

The hypersaline Solar Lake in Taba, Egypt, is a heliothermal lake (12). The lake is 140 m × 70 m and is 4–6 m deep (11). The Solar Lake undergoes seasonal salinity stratification; in summer, holomixis occurs due to evaporation, the lake water becomes completely oxic, and the salinity can reach 18% (11). In fall, the existing highly saline water and the introduction of fresh seawater from the Red Sea create a salinity gradient and water stratification (10, 11). The effect of the limnological cycle on the Solar Lake microbial communities remains largely unexplored. The lake is known to harbor a 1-mm thick lithifying cyanobacterial mat, and it is considered among the few mat ecosystems that are analogous to Precambrian stromatolites (7, 8). Studies on the Solar Lake benthic microbial communities ceased in the late 1990s, leaving a significant knowledge gap (7, 8, 13–16). Earlier studies on the Solar Lake benthic microbial communities indicated that there are four types of mats, including the shallow-water flat mat surrounding the lake (8, 17). Between the late 1970 and 1990s, studies on shallow-water flat microbial mat showed the prevalence of *Cyanobacteria* belonging to *Coleofasciculus chthonoplastes* (previous name: *Microcoleus chthonoplastes*), *Oscillatoria salina*, *Oscillatoria limnetica*, *Spirulina labyrinthiformis*, *Spirulina* sp., and *Aphanothece halophytica*. Sulfate reducers belonging to *Desulfovibrio* and *Desulfonema* along with methanogens thrived in the shallow-water flat mat (13, 16). The Solar Lake benthic methanogens were enriched in the presence of methylated amines and $H_2/CO_2$ rather than acetate (13).

Recent studies using bacterial 16S rRNA gene sequencing of the Solar Lake sediment showed that 72% of the sediments' bacterial community were unclassified at the genus level (18, 19). As expected, sulfur oxidizing and sulfate-reducing communities were abundant in the lake sediments (18, 19). The lake limnological cycle has been shown to affect the shallow-water microbial mat (8). During the summer season and due to the high evaporation rate, the water recedes partially from the flat shallow-water mat, which becomes exposed (8, 20). The exposed mat is likely to undergo heightened salinity and increased oxygen infiltration due to drought, as observed in similar ecosystems (21, 22). Studying the microbial communities in the Solar Lake shallow-water versus exposed mats can provide valuable insights into the mechanisms of adaptation employed by microbial communities to cope with environmental stressors caused by the reduction in summer water level.

In this study, we revisited the Solar Lake flat microbial mats during the summer of 2021 to reveal the identity and functional potential of the benthic archaeal and bacterial communities with a predominant interest in understanding the genetic adaptation of the exposed versus shallow-water community to summer partial water recession, specifically phototrophic and methanogenic communities. We anticipated that the microbial community within the flat mats would experience a change in the genetic makeup causing taxonomic and biochemical zonation in the exposed versus shallow water in response to summer exposure. Hence, we sampled from both shallow water and exposed mat sites and applied in-depth sequencing using Illumina Novaseq followed by extensive analyses of raw reads, metagenome assembly, and metagenome-assembled genomes (MAGs) binning. Additionally, the absolute abundance of bacterial, archaeal, methanogenic, and sulfate-reducing communities was investigated.

## MATERIALS AND METHODS

### Sample collection and measuring mats physicochemical parameters

Triplicate sediment samples were collected from two shallow water and two exposed flat microbial mats in July 2021 (Fig. S1). The N-SO$_{shallow-water}$ and N-SO$_{exposed}$ samples were collected from sediments in the northern mountainside region covered by a small mountain shade during field sampling (12:00–13:00), and the E-SO$_{shallow-water}$ and E-SO$_{exposed}$ samples were collected from sediments in the eastern seaside part of the Lake, exposed to sunlight and closer to the Red Sea (Fig. S1). The shallow-water mat was covered by ~10 cm of lake water.

Samples were collected using handmade cores (5 cm diameter and 10 cm length) and were kept at 4°C until they reached the American University in Cairo (AUC) laboratories. The cores were made of polypropylene 50 mL syringe, cut after the tip by the university workshop, and autoclaved before use. The sediment cores were homogenized and then separated into 1.5 mL screw-capped tubes, under sterile conditions at AUC, then shock-frozen for later processing. The homogenized samples are referred to as "mats" throughout the paper.

The Hanna Combo pH/EC/TDS/Temp tester (Model HI98129) (Hanna instrument, Woonsocket, RI, USA) was used to measure pH and temperature in the field.

A subset of the homogenized sediments was dried at 60°C for 72 h (according to MEDAC LTD instructions). Each dried replicated sediment sample was equally mixed into one composite sediment and sent to MEDAC LTD (Chobham, UK) for total carbon, hydrogen, nitrogen, sulfur, and anions (sulfate, nitrate, and nitrite) quantification. The sediments' gravimetric water contents were measured by drying ~1 g at 60°C for 72 h (23).

### DNA extraction and NGS sequencing

Frozen sediments (0.2–0.4 g) were washed three times with phosphate buffer saline (PBS) buffer (ratio 1:5) and centrifuged at 3,000 RPM for 3 min. Washed sediment samples were then treated with lysozyme (100 mg/mL) for 1 h at 37°C, followed by proteinase K (20 mg/mL) for 30 min at 56°C. Treated samples were then processed with the DNeasy PowerSoil Kit (Qiagen, Hilden, Germany). The DNeasy PowerSoil Kit was used per the manufacturer's protocol, except for an hour bead beating step using a vortex adapter.

Solar Lake shallow-water and exposed microbial mats were sequenced in triplicate at Novogene (Cambridge, UK) using the Illumina NovaSeq 6000 (PE150). Each replicate had a sequencing depth of 20 M reads (Table S1).

### Absolute abundance of bacterial, archaeal, methanogenic, and sulfate-reducing communities

Gene copies (qPCR) of bacterial and archaeal (16S) rRNA, *dsrB,* and *mcrA* genes were quantified using the Sybr Green-based assays as previously described (24, 25). The calibration curve for 16S rRNA and *mcrA* genes was created as previously described (24), whereas the *dsrB* gene calibration curve was made using a PCR amplicon of *dsrB* from *Desulfovibrio vulgaris* (10–10$^5$ *dsrB* copies). Microbial absolute abundance quantification was carried out using CFX Connect Real-Time PCR detection system (Bio-Rad, CA, USA). qPCR reaction efficiency was at least 80% ($R^2 > 0.98$). Melting curve analysis was used to assess the presence of unspecific products.

### Metagenomes, assembly, binning, and MAGs refinement

Quality control of the raw reads was performed using fastp (version 0.23.2) default parameters (26). High-quality paired-end reads were taxonomically classified using Kaiju (version 1.9.2) and the nr database (27).

High-quality reads from the triplicate sequencing files were co-assembled using Megahit (version 1.2.9) with minimum kmer 27, maximum kmer 127, and 10 kmer step

(28). Individual replicate files from each sample were mapped to the sample's assembled contigs using BBMap version 38.94 (options minid = 0.9, covstats, and scafstats were used) (29). MAGs were binned using MetaBAT2 (Version 2.12.1) (30). CheckM2 (version 1.0.0) was used to examine MAGs' quality (completeness and contamination levels) (31). Any MAG failing the SAG/MAG current community standards (32), with >10% contamination, was refined by mdmcleaner (version 0.8.3) (33) and then subjected to another round of quality checks using CheckM2 (31). All high-quality (>70% completeness & <5% contamination) and medium-quality (>50% completeness and < 10% contamination) bins were further analyzed.

## MAGs taxonomic classification, phylogeny, relative abundance, beta-diversity, and PTR calculation

The genome taxonomy database (GTDB) was used to taxonomically classify MAGs generated from each metagenome as a part of the mdmcleaner pipeline (33, 34). The CheckM utility command "profile" was used to assess the MAGs' relative abundance within each metagenome; the data presented are based on the percent community estimation (35).

The draft MAGs maximum likelihood phylogenomic tree was generated using the GTOTree workflow (version 1.7.05) based on the alignment of 25 archaeal and bacterial single-copy housekeeping genes (SCGs) (36). Prodigal (version 2.6.3) was used to predict the open reading frames (ORFs), and then, target SCGs were identified with HMMER3 (version 3.3.2) (37, 38). SCGs were then individually aligned with muscle (version 5.1), trimmed with trimal (version 1.4.1), and FastTree2 (version 2.1.11) was used to concatenate SCGs and perform the phylogenetic tree (39–41). Only 227 out of 292 MAGs are displayed on the tree, since 69 MAGs contained few SCGs or had >10% redundancy with one or more represented MAG/s.

A Solar Lake archaea-specific tree was created to understand the relation of our archaeal MAGs with previously described taxa, based on 16S rRNA gene fragments from water samples from the Solar Lake (42). The CheckM utility command "ssu finder" was deployed on all archaeal MAGs to retrieve the 16S rRNA gene, when applicable (35). Water-column archaeal 16S rRNA partial genes were retrieved from the GeneBank database (43). Cytryn et al. (42) was the only study that analyzed the Solar Lake archaeal 16S rRNA, to date (42) . Water-column and MAGs 16S rRNA genes were aligned with muscle and manually trimmed (39, 41). The Maximum likelihood tree was generated using PhyML using 1,000 bootstraps (44). Trees were visualized using the Interactive Tree Of Life (iTOL) server (45). Beta-diversity analysis of shallow-water and exposed prokaryotic community and Compute Peak-to-Trough ratio (PTR) of MAGs calculation methods are presented in the supplementary file. Additionally, codes and scripts that were used to generate the metagenomic assemblies and MAG annotation are included in the supplementary file.

## Metagenomes assemblies and MAGs annotations

Metabolic reconstruction of metagenome contigs and MAGs was carried out using the Distilled and Refined Annotation of Metabolism (DRAM, version 1.4.3) tool (46). MAGs' ORFs were called using Prodigal (version 2.6.3) and were annotated using KOfam, Uniref90, Pfam, and dbCAN databases (access date: 4 January 2023) (47–50). The metagenome contigs were annotated using the KOfam, pfam, and dbCAN databases (48–50). All annotations presented in this study were retrieved from either distillate or product files generated by DRAM, except the environmental adaptation gene annotation which were retrieved from the DRAM raw annotation file. Genes would be considered correctly annotated if detected by at least two out of four databases.

For each sample, contigs coverage was extracted from the covstats files generated by BBMap. Contig coverage was used as an indication of gene copy numbers. The count per million (CPM) of functional genes, pathways, and Kyoto Encyclopedia of Genes

and Genomes (KEGG) modules presented in the study was calculated according to the following formula for each replica:

$$\text{Gene CPM} = \frac{\text{contig coverage}_x}{\text{total number of reads}} \times 10^6 \text{genes per million reads}$$

Where contig coverage$_x$ is the contig coverage of all contigs containing the functional gene in the sample.

## Statistical analysis

The Kruskal–Wallis test was applied to all qPCR data (51). Followed by an ad-hoc multiple comparison adjustment/correction test (Dunn test) (52). Deseq2 differential abundance pipeline was applied on N-SO$_{\text{shallow-water}}$ versus N-SO$_{\text{exposed}}$ and E-SO$_{\text{shallow-water}}$ versus E-SO$_{\text{exposed}}$ pairs to infer the significant difference between taxonomic assignments of exposed versus shallow-water microbial mats (53). The significant difference in the functional pathways, modules, and genes CPM between the exposed and shallow-water mats was tested using an unpaired-t-test followed by a two-stage step-up $P$-value false discovery rate (FDR) correction using the Benjamini, Krieger, and Yekutieli methods (54).

## RESULTS

### Physicochemical characteristics of Solar Lake microbial flat mats

The exposed and shallow-water microbial mats had neutral pH (6.78–7.63), except for N-SO$_{\text{exposed}}$, which was mildly alkaline (pH 8.00). The temperature ranged between 35.26°C and 38.00°C in the shallow-water mats and between 39.73°C and 39.86°C in the exposed mats (Table 1). As expected, the water content in the exposed (31%–28%) was lower than the shallow-water mats (54%–55%). Nitrogen content ranged between 0.29% and 0.31%. Nitrate concentration was the lowest in the N-SO$_{\text{exposed}}$ mat (100 ppm) (Table 1).

### Metagenome-driven taxonomic and functional annotation of Solar Lake shallow-water and exposed flat mat microbiome

The raw reads and the assembled metagenomes were annotated to deduce the taxonomic distribution and identify specific potential functional characteristics, respectively. The bacterial community in the Solar Lake mats had an absolute abundance ranging between 2.34E+10 (N-SO$_{\text{shallow-water}}$) and 1.32E+11 (E-SO$_{\text{shallow-water}}$), whereas archaeal absolute abundance ranged between 2.72E+09 (N-SO$_{\text{shallow-water}}$) and 8.38E+10 (E-SO$_{\text{shallow-water}}$) (Fig. 1). Taxonomic classification of the raw reads showed that the archaeal microbial community was mainly composed of *Euryarchaeota*, Asgardarchaeota, *Candidatus* Thermoplasmatota, *Bathyarchaeota,* and members of the DPANN (acronym of Diapherotrites, Parvarchaeota, Aenigmarchaeota, Nanohaloarchaeota, and Nanoarchaeota). Meanwhile, the bacterial community was mainly composed of *Pseudomonadota* (Proteobacteria), *Chloroflexota*, *Bacteroidota,* and *Cyanobacteria* (Fig. S2). Raw reads taxonomic classification agrees with the Solar Lake MAGs taxonomic assignments (Fig. 1B and C).

**TABLE 1** Physicochemical characteristics of Solar Lake sediments

| | Temperature (C°) | Water content (%) | pH | H % [a] | C % [a] | N % [a] | S % [a] | Nitrite (ppm)[a] | Nitrate (ppm)[a] | Sulfate (ppm)[a] |
|---|---|---|---|---|---|---|---|---|---|---|
| N-SO$_{\text{shallow-water}}$ | 35.26 (±0.25) | 55 (±10) | 7.61 (±0.28) | 4.52 | 1.09 | 0.31 | 1.05 | 900 | 3,400 | 7,500 |
| N-SO$_{\text{exposed}}$ | 39.86 (±1.36) | 31 (±2) | 8.00 (±0.00) | 2.80 | 1.13 | 0.28 | 0.77 | 500 | 100 | 9,400 |
| E-SO$_{\text{shallow-water}}$ | 38.00 (±0.82) | 54 (±11) | 6.78 (±0.27) | 5.15 | 1.09 | 0.29 | 0.79 | 100 | 2,300 | 30.800 |
| E-SO$_{\text{exposed}}$ | 39.73 (±0.87) | 28 (±3) | 7.63 (± 0.26) | 3.21 | 1.00 | 0.29 | 1.14 | 100 | 1,700 | 11,900 |

[a]per gram dry weight.

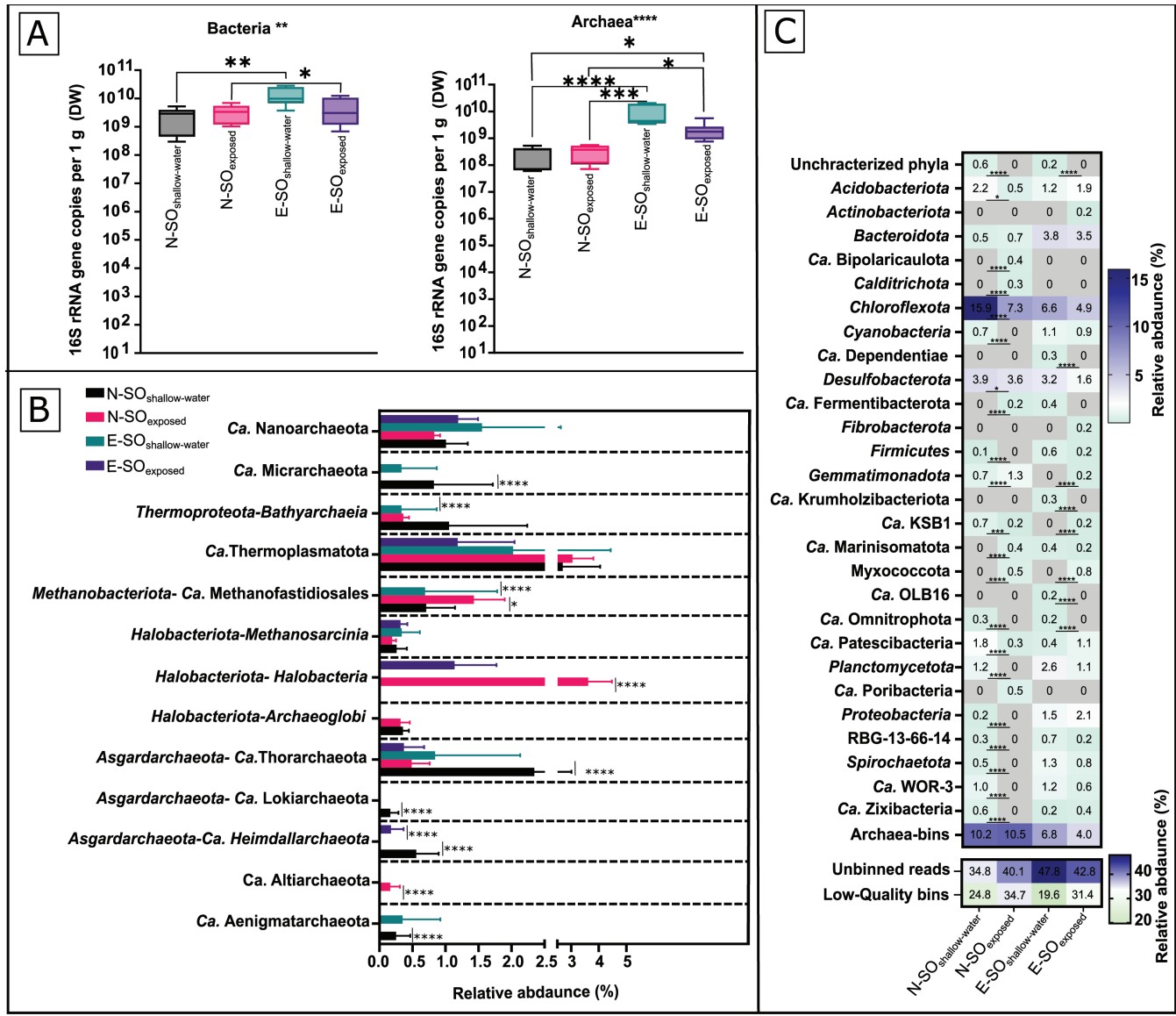

**FIG 1** Relative and absolute abundance of the Solar Lake microbial shallow-water and exposed benthic communities. (A) Archaea and bacteria absolute abundance based on 16S rRNA gene copy numbers per gram dry weight. (B) Archaeal MAGs relative abundance. (C) Bacterial MAGs relative abundance. The relative abundance of MAGs reported is based on the proportion of a bin relative to the number of reads mapped to assembled contigs and adjusted for the size of the bin (CheckM "% community"). $P$-values indicate the statistical difference between samples ($P$-value $\leq 0.05$ = *, $P$-value $\leq 0.01$ = **, $P$-value $\leq 0.001$ = ***, $P$-value $\leq 0.0001$ = ****). In panels B and C, DESeq2 was used to infer the significantly different abundant phyla between each exposed and shallow-water sample pair (N-SO_shallow-water versus N-SO_exposed and E-SO_shallow-water versus E-SO_exposed). For qPCR data, Kruskal–Wallis test was used for the analysis of variance between samples (depicted above the qPCR graph), and the Dunn test was used for multiple comparison correction ($P$-values are depicted above the data points).

Annotations of the Solar Lake assemblies were performed to determine the genetic capacity of the microbial communities in the exposed and shallow-water mats (Fig. 2A and B; Table S2).

Genes responsible for photosystem I (*psaA*) and II (*psbA* and *psbD*) and anoxygenic photosystem II (*pufL* and *pufM*) were detected in all samples. The gene content of the Solar Lake mats is suggestive of autotrophic communities that depend on the Wood-Ljungdahl cycle (40–65 CPM), Arnon-Buchanan cycle (20–26 CPM), and Calvin cycle (5–11 CPM) (Fig. 2A).

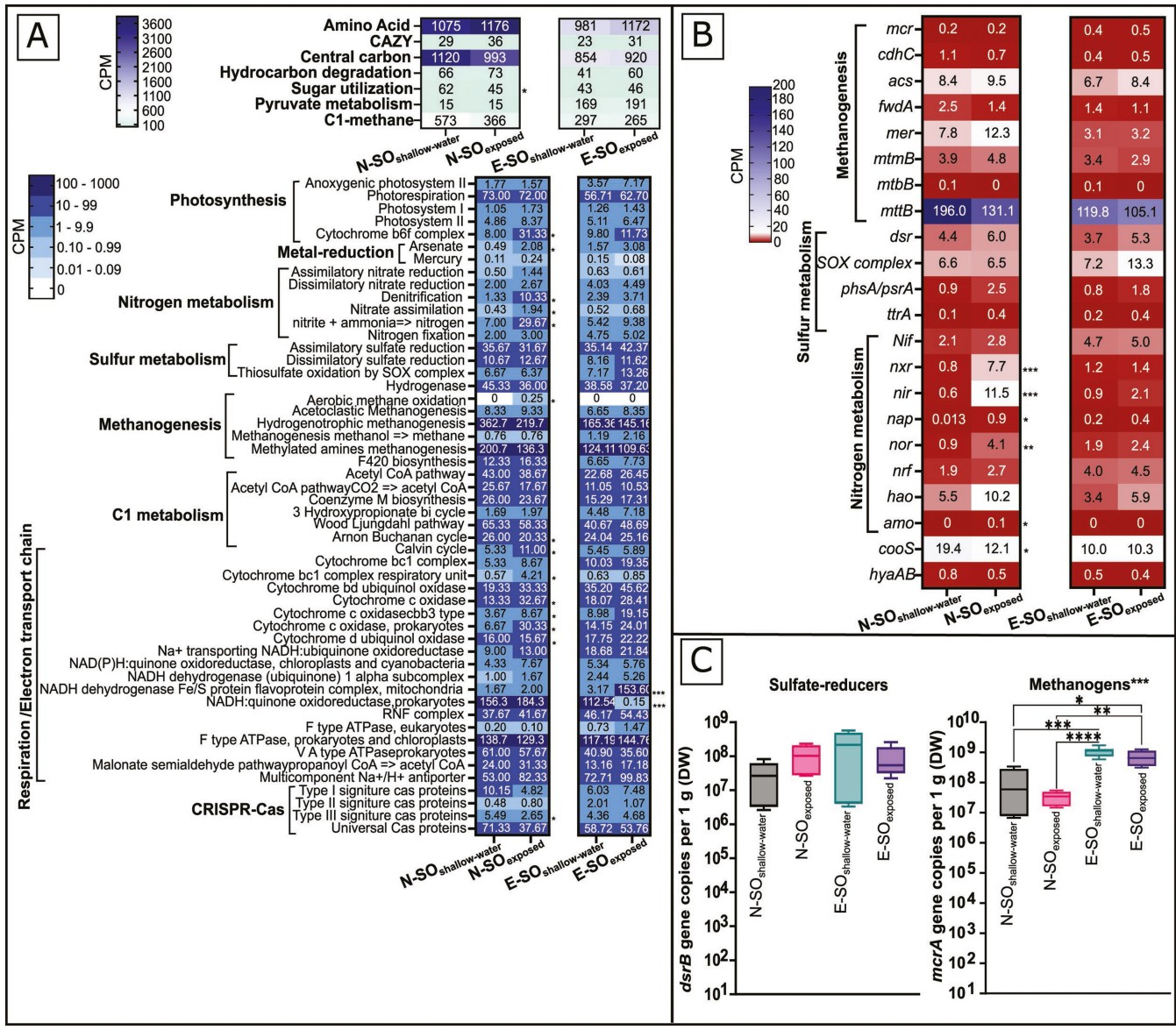

FIG 2 Distribution of functional pathways, KEGG modules, and key functional genes across the shallow-water and exposed Solar Lake microbial mats metagenome assemblies. (A) Heat map showing the CPM per functional pathway or KEGG module. (B) Heat map showing CPM of the key functional genes involved in methanogenesis, sulfur, and nitrogen metabolism, in the assemblies. Numbers shown in each cell represent CPMs for each pathway, module, or functional gene. (C) Box plots showing the number of gene copies per gram dry weight (DW) of *mcrA* and *dsrB*, respectively. *P*-values (FDR corrected) indicate statistical differences between exposed and shallow water communities (*P*-value ≤ 0.05 = *, *P*-value ≤ 0.01 = **, *P*-value ≤ 0.001 = ***, *P*-value ≤ 0.0001 = ****). In panel C, the Kruskal–Wallis *P*-values are depicted above the graphs, whereas Dunn test *P*-values are depicted between each significantly different pair of data.

The relative abundance of genes classified by the CAZy database responsible for the degradation of complex carbohydrates ranged between 29 and 36 CPM (Fig. 2A). The AA1 (laccases/ferroxidases) carbohydrate-active enzyme (Cazyme) family was significantly enriched (10 CPM, *P*-value < 0.05) in the N-SO$_{exposed}$ mat (Fig. S3).

The genes encoding for the methanogenesis key enzyme, methyl coenzyme M reductase (*mcrAB*), were detected in all samples. The *mcrAB* had comparable abundance (CPM) (Fig. 2B) with the *mcrA* absolute abundance (qPCR) (Fig. 2C), with N-SO$_{shallow-water}$ and N-SO$_{exposed}$ pair having lower CPM compared with E-SO$_{shallow-water}$ and E-SO$_{exposed}$ (Fig. 2B and C). The four Solar Lake metagenomes had the key genes involved in the production of methane from acetate (*cdhc* and *acs*), monomethylamine

($mtmB$), trimethylamine ($mttB$), and $H_2/CO_2$ ($mer$ and $fwdA$) (Fig. 2B). The genetic potential for aerobic methane oxidation was detected in the N-SO$_{exposed}$ sample only (Fig. 2A).

The solar lake shallow-water and exposed microbial mats have the genetic potential to reduce sulfate as well as tetrathionate and oxidizing/reducing thiosulfate (Fig. 2A). The $dsrB$ gene, encoding sulfate reduction, had a similar relative (CPM) and absolute (qPCR) abundance (Fig. 2B and C). All Solar Lake mat samples had a similar nitrogen fixation key gene abundance ($nif$). In contrast, genes involved in nitrification ($amo$ and $hao$) and denitrification ($nir$) were enriched ($P \leq 0.05$ and 0.001, respectively) in N-SO$_{exposed}$ (Fig. 2A and B).

## An overview of bacterial and archaeal MAGs

A total of ~240 M high-quality raw reads (~60 M for each metagenome) were assembled and binned to generate a total of 108 archaeal and 601 bacterial MAGs (Table S1). This study investigated predominantly medium-to-high-quality MAGs, which included 72 archaeal and 220 bacterial MAGs (Table S3), with the exception of eight archaeal MAGs with ≥50% completeness and ≤10% contamination criteria according to mdmcleaner, and not CheckM2, and the only *Candidatus* Altiarchaeota MAG (S-SO2-bin.64) with 49% completeness (Table S3)

MAGs summary, including but not limited to MAGs completeness, contamination, GTDB taxonomic assignment, and the number of predicted genes, is presented (Table S3).

The Solar Lake MAGs spanned nine archaeal and 28 bacterial phyla based on GTDB taxonomy (Fig. 3; Table S3). The archaeal MAGs assigned to *Candidatus* Thermoplasmatota had the highest relative abundance in all the samples (Fig. 1B). We also detected MAGs belonging to the deeply branching Asgardarchaeota superphylum in all mats.

MAGs assigned to *Candidatus* Aenigmatarchaeota, and *Candidatus* Micrarchaeota were shallow-water microbial mat-specific MAGs. On the contrary, MAGs assigned to *Halobacteriota* (class: *Halobacteria*) were only detected in the exposed microbial mat (Fig. 1B; Table S3).

Shallow-water flat mat-specific bacterial MAGs were assigned to uncharacterized phyla (2 MAGs) and *Candidatus* Omnitrophota (2 MAGs), and MAGs assigned to *Myxococcota* were detected only in the exposed mats (Fig. 1C).

MAGs and reads assigned to the *Chloroflexota* phylum had the highest abundance across all samples (Fig. 1C; Fig. S2). Sulfate-reducing bacteria assigned to the *Desulfobacterota* phylum represented 1.6-3.9% of the total Solar Lake benthic community (Fig. 1C). Cyanobacterial MAGs had a relative abundance of 0.7%–1.1%.

## Phototrophic potential of Solar Lake MAGs

We detected genes encoding for phototrophy in a total of 17 Solar Lake microbial mat MAGs, with six cyanobacterial and 11 anoxygenic phototrophic MAGs. Three of the six cyanobacterial MAGs belong to *C. chthonoplastes* (Fig. 4A and B; Table S3). The highest relative abundance for *C. chthonoplastes* was detected in E-SO$_{exposed}$ (~0.87%), followed by N-SO$_{shallow-water}$ (~0.56%) and E-SO$_{shallow-water}$ (0.39%) (Table S4). The genes encoding photosystem I and II in the *C. chthonoplastes* MAGs showed higher CPMs by trend than other photosynthetic *Cyanobacteria* (Fig. 4A). *C. chthonoplastes* MAGs possessed a nearly complete set of genes for the Calvin cycle and had the genetic potential to degrade amorphous cellulose, fix nitrogen ($nifH$), and produce lactate, acetate, and alcohol (Fig. 4B).

The other cyanobacterial MAGs were only detected in shallow-water mats (Fig. 4A and B; Table S4) and included an *Elainellaceae* MAG (National Center for Biotechnology Information [NCBI] taxonomy: unclassified *Leptolyngbya*) assembled from the N-SO$_{shallow-water}$ metagenome and *Oscillatoria* and *Halothece* unclassified species from the E-SO$_{shallow-water}$ (Fig. 4A and B; Table S4). Similar to *C. chthonoplastes,* shallow-water mats cyanobacterial MAGs encoded a nearly complete set of genes for the Calvin cycle

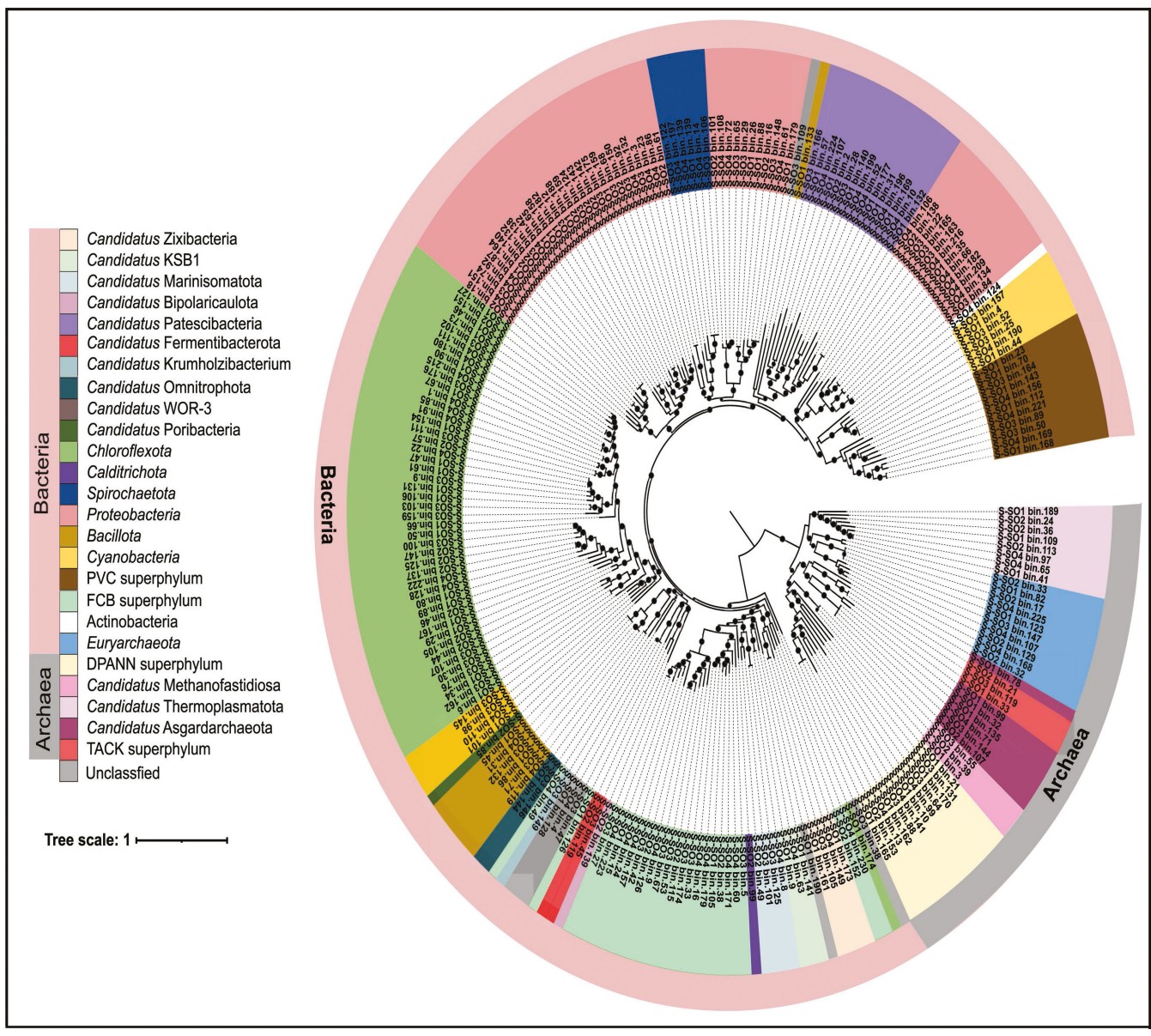

**FIG 3** Phylogenomic tree of the Solar Lake shallow-water and exposed mat MAGs. The maximum likelihood phylogenomic tree of up to 25 archaeal and bacterial single-copy housekeeping genes (SCGs) was generated using the GoToTree workflow. Nodes with bootstrap higher than 50% are represented by a closed circle. MAGs assigned to different superphyla or phyla when no superphylum is available are highlighted by different colors. MAGs assignment to a superphylum or phylum was based on GTDB taxonomic assignment.

(Fig. 4B). The *Oscillatoria* MAG additionally showed the genetic potential to fix nitrogen and degrade amorphous cellulose (Fig. 4B).

Of the 11 anoxygenic phototrophic MAGs, only one MAG (S-SO2-bin.152) showed the genetic potential for anoxygenic phototrophy in the N-SO$_{exposed}$ sample and was taxonomically assigned to an unclassified class within *Gemmatimonadetes*. The former had genes involved in the Arnon-Buchanan cycle for carbon fixation and alcohol and acetate production (Fig. 4B). In E-SO$_{shallow-water}$, two MAGs assigned to an unknown genus within *Chloroflexaceae* had the genetic potential for phototrophy, carbon fixation via the Wood-Ljungdahl pathway, and the ability to fix nitrogen (Fig. 4B).

Seven of the 11 MAGs belonged to the *Alphaproteobacteria*. Five of these MAGs contained genes for the 3-hydroxypropionate bi-cycle for carbon fixation (Fig. 4B), whereas six MAGs had the genetic capacity for thiosulfate oxidation. One MAG assigned

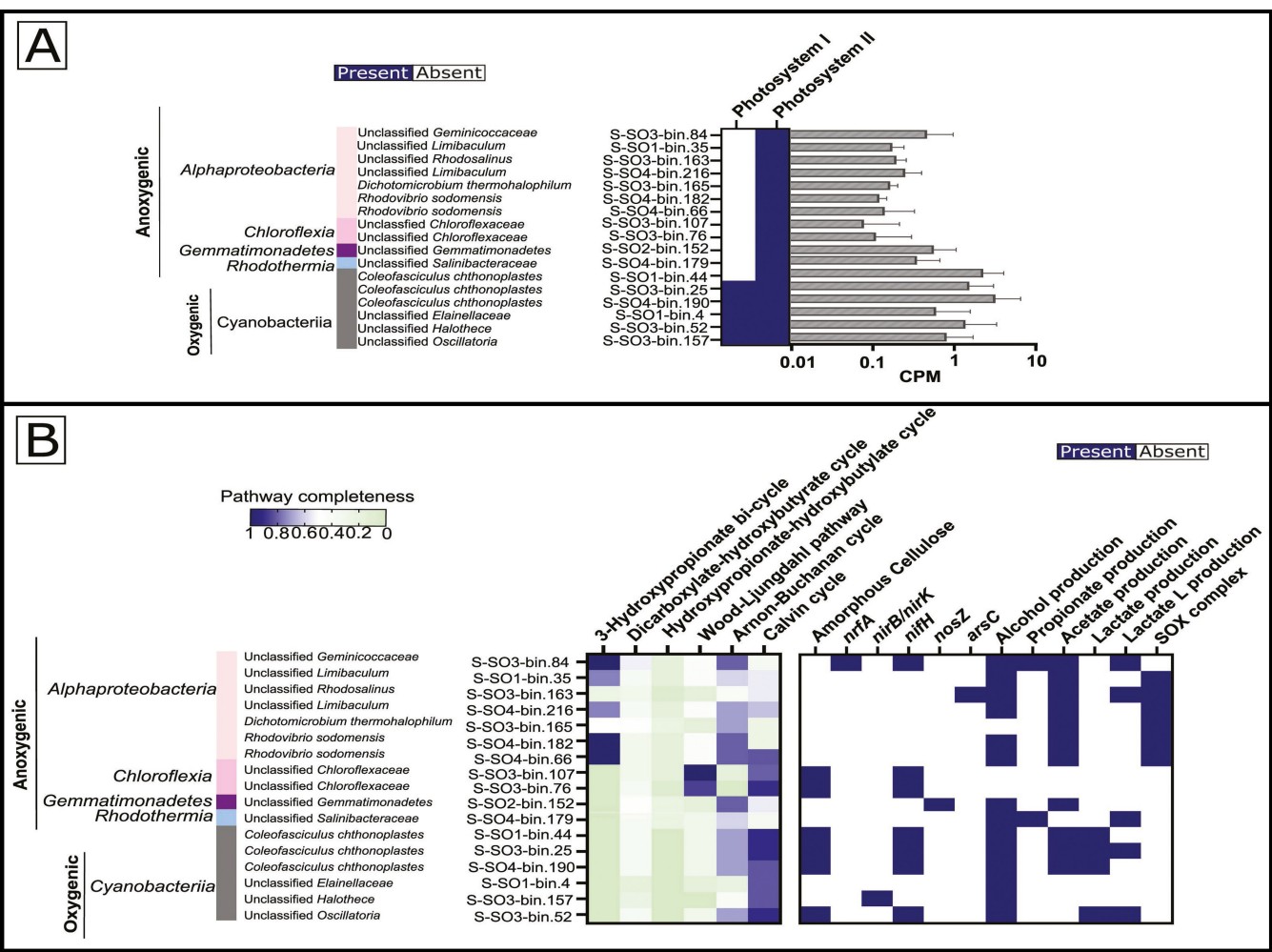

**FIG 4** The functional genetic potential of the recovered phototroph MAGs in the Solar Lake mats. (A) Photosystem I and II in phototrophic bacterial MAGs, (B) Carbon fixation and other metabolic pathways in phototrophic MAGs. The genetic potential for phototrophy in MAGs is indicated by the presence or absence and the CPM of photosystem I and II. Carbon fixation pathways are indicated by percent completeness in the Solar Lake MAGs. One indicates the detection of a complete pathway.

to an unknown genus within the *Geminicoccaceae* family had the genetic potential to reduce nitrite to ammonia (*nrfA*) and fix nitrogen (*nifH*).

## Methanogenesis potential of Solar Lake MAGs

Genes encoding the methanogenesis pathway were detected in different archaeal MAGs. A complete methanogenesis pathway was detected in four nearly complete MAGs across the four Solar Lake mat samples (Fig. 5A). These MAGs were taxonomically classified within the *Methanosarcinia* class (species: *Methanohalobium evestigatum*). All *M. evestigatum* MAGs had the key genes responsible for acetolactic (*acs*), methyleamine (*mtmB*), and methanogenesis (Fig. 5A).

Additionally, more than half of the genes involved in acetolactic methanogenesis were detected in three MAGs that belonged to the *Candidatus* Methanofastidiosales (Fig. 5A). These three MAGs showed the genetic potential to degrade amorphous cellulose, fix nitrogen, and oxidize thiosulfate (Fig. 5B).

Moreover, three MAGs affiliated with the *Bathyarchaeia* class encoded a nearly complete set of methanogenesis genes (Fig. 5A). The functional annotation analysis suggests that the microorganisms represented by these MAGs are capable of producing

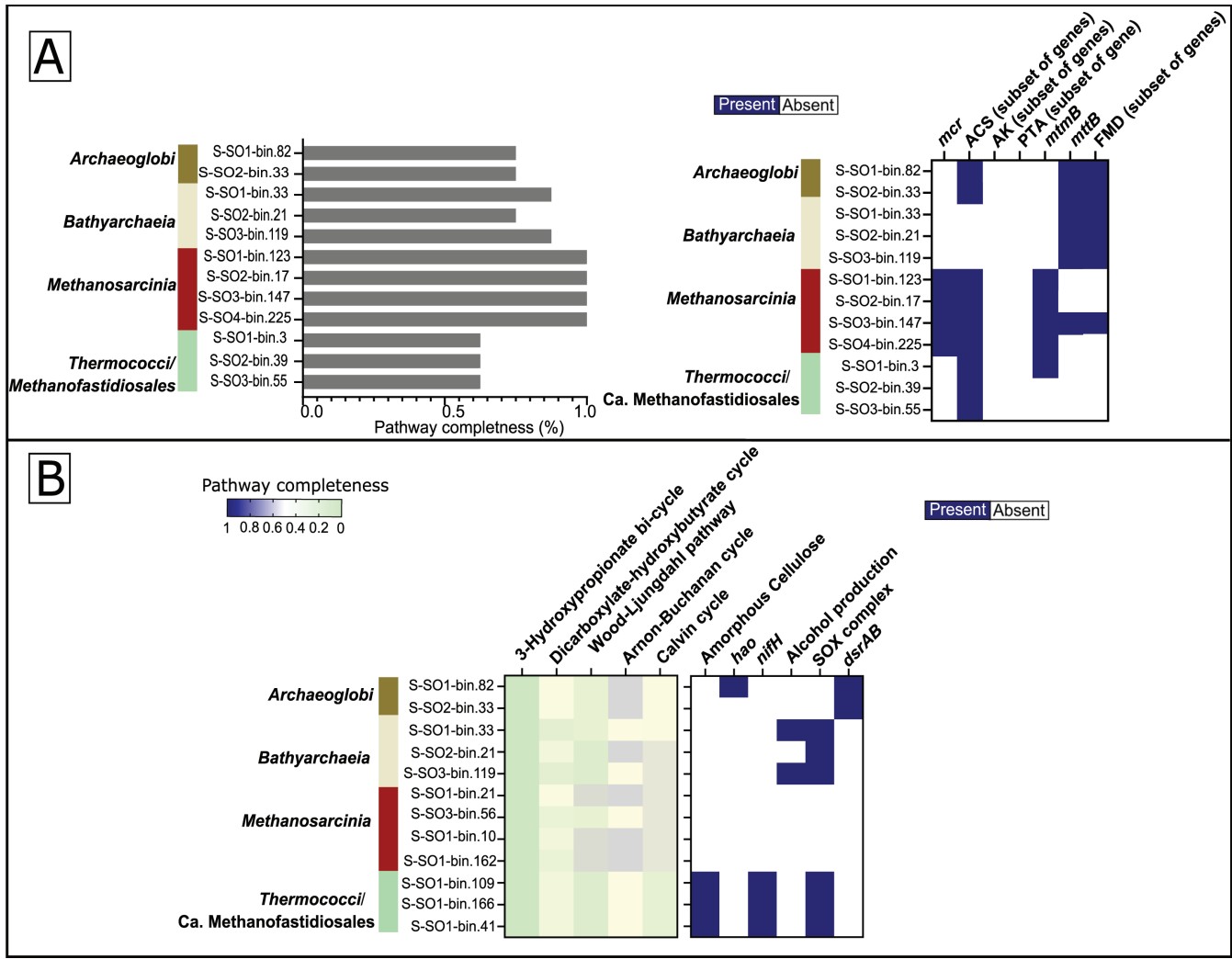

**FIG 5** The functional genetic potential of the recovered methanogen MAGs in the Solar Lake mats. (A) Methanogenesis pathway completeness and key functional genes in the Solar Lake MAGs. (B) Carbon fixation and selected metabolic pathway key genes in the putative methanogen MAGs. The genetic potential of each MAG to produce methane is indicated by methanogenesis pathway completeness and the presence of key methanogenesis. Carbon fixation and methanogenesis pathways are indicated by percent completeness. One indicates the detection of a complete pathway.

methane from trimethylamine (*mttB*) and $CO_2$ (FMD) (Fig. 5A). Besides methanogenesis, the *Bathyarchaeia* MAGs had the genetic potential to oxidize thiosulfate and produce alcohol.

In addition, two *Archaeoglobaceae* MAGs had most of the genes involved in acetoclastic and trimethylamine-driven methanogenesis. These two *Archaeoglobaceae* MAGs also showed the genetic potential to perform dissimilatory sulfate reduction (Fig. 5A and B).

## Differential ROS detoxification, UV radiation resistance, and salt-tolerance genes in the Solar Lake microbial shallow-water versus exposed mats

Both the Solar Lake shallow-water and exposed microbial mats' metagenomes and MAGs encoded salt tolerance genes such as the osmoprotectant transport systems (*opuABCD*), the glycine betaine/proline transporter system (*proV*), and the betaine-aldehyde dehydrogenase (*betB*). Additionally, the relative abundance (CPM) of the genes encoding salt tolerance was significantly greater ($P \leq 0.05$) in the N-SO$_{exposed}$ than in the N-SO$_{shallow-water}$ metagenome (Fig. 6A). All 17 MAGs defined by their genetic

potential for phototrophy had at least one copy of these osmoprotectant genes; except S-SO1-bin.4 and S-SO4-bin.179. On the contrary, within the 12 methanogenic archaeal communities, osmoprotectant genes were detected only in four *M. evestigatum* MAGs and one *Archaeoglobaceae* MAG (Fig. 6B).

Compared with N-SO$_{shallow-water}$, the relative abundance (CPM) of ROS detoxifying genes, such as catalase-peroxidase (*katE*) and superoxide dismutase genes (SOD1) in the metagenome, was significantly greater ($P \leq 0.05$ and 0.01) in N-SO$_{exposed}$ (Fig. 6A) compared with N-SO$_{shallow-water}$. Additionally, the E-SO$_{exposed}$ had higher CPM of ROS detoxifying genes compared with E-SO$_{shallow-water}$, but only by trend (Fig. 6A). Note that all the MAGs representing phototrophic populations had at least one or more genes involved in ROS response (Fig. 6B).

The *urvD* and the *RuvAB* genes encoding UV radiation resistance genes were detected in all 17 phototrophic MAGs defined by their genetic potential for phototrophy, except S-SO1-bin.4 and S-S03-bin.165 (Fig. 6B). *RuvAB* and *urvD* were absent in all MAGs affiliated with methanogens.

## DISCUSSION

The heliothermal Solar Lake in Taba, Egypt, is expected to host distinctive microbial communities able to adapt to the lake's limnological cycle (11). Our present study is the first to revisit the Solar Lake benthic microbial communities since 1998 (16). We used a genome-centric approach to highlight the structural and functional adaptations occurring in exposed versus shallow-water mats in response to the summer's partial water receding. We focused on the cyanobacterial and methanogenic communities. This study is a logical continuation of previous reports from 1970s and 1980s, which relied on conventional microbiology techniques (8, 13).

Bacterial and archaeal absolute abundance was comparable in the exposed (N-SO$_{exposed}$ and E-SO$_{exposed}$) versus shallow water (N-SO$_{shallow-water}$ and E-SO$_{shallow-water}$) (Fig. 1). Unlike the absolute abundance, archaeal and bacterial phyla-relative abundance and beta-diversity significantly shifted between the exposed and shallow-water flat microbial mats (Fig. 1; Fig. S4).

### Archaeal benthic community in the shallow versus exposed Solar Lake microbiome

MAGs assigned to the symbiotic *Candidatus* Micrarchaeota and *Candidatus* Aenigmarchaeota were uniquely detected in the shallow-water microbial mats. It is unclear why *Ca.* Aenigmarchaeota and *Bathyarchaeia* (order B26-1) MAGs were specifically detected in the shallow-water microbial mat (S-SO1_bin.160 and S-SO3_bin.10). However, horizontal gene transfer events between the two phyla were previously reported, hence emphasizing the symbiotic relationship between *Bathyarchaeia* and *Ca.* Aenigmarchaeota members (55, 56). Both *Ca.* Aenigmarchaeota and *Bathyarchaeota* were detected in other microbial mats (57), deep-sea (56) freshwater sediments (58), coastal sediments (59), and groundwater sediments (60).

The aerobic heterotrophic *Halobacteria* class (1), previously reported to be the most abundant archaeal member in the Solar Lake water during summer holomixis (42), was not detected in the shallow-water microbial mats community (Fig. 1B). This could be driven by the high salinity and density of the water creating a sub-oxic/anoxic environment disfavoring *Halobacteria* growth (11, 18). On the contrary, *Halobacteria* MAGs were enriched in the Solar Lake exposed samples (N-SO$_{exposed}$ and E-SO$_{exposed}$) (Fig. 1B) and denitrifying, possessing *nir* gene, *Halobacteria* (2.5%) and *Anaerolineae* (3.3%) MAGs were only enriched in N-SO$_{exposed}$.(Table S5). The latter bacterial genus is included because it has been previously shown to be involved in denitrification in hypersaline ecosystems (61–64). Nevertheless, *Anaerolineae* has not been associated with nitrite reduction and nitric oxide generation (*nir* genes) (64). The relative depletion of both nitrate and nitrite in N-SO$_{exposed}$ (Table 1) supports the presence of active denitrification.

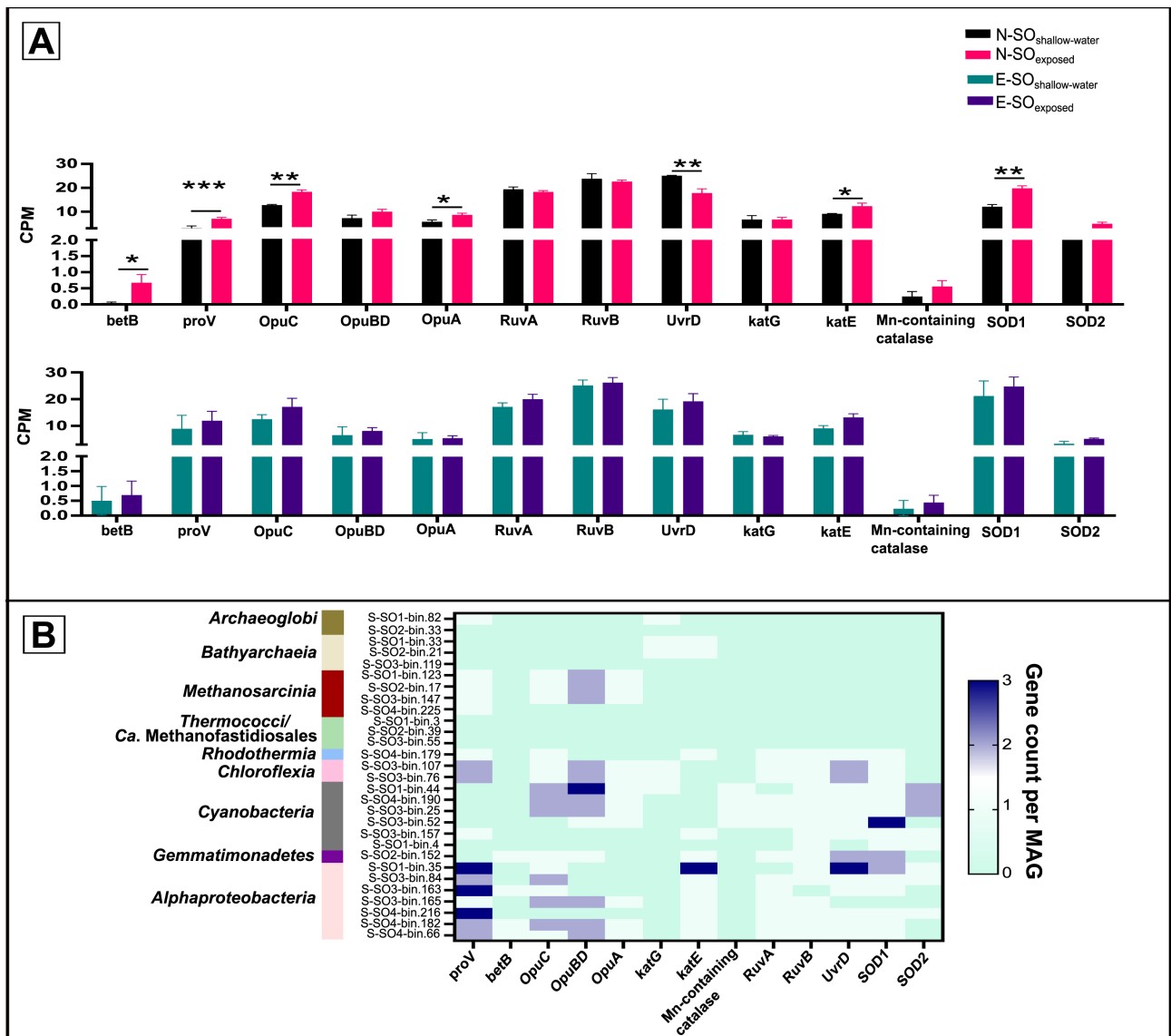

**FIG 6** Environmental adaptation genes in the Solar Lake shallow-water and exposed microbial mats. (A) Relative metagenomic abundance (CPM) of genes encoding stress tolerance. (B) Distribution pattern (CPM) of stress-tolerance genes among MAGs representing phototrophic or methanogenic populations. $P$-values (FDR corrected) are shown at the top of each comparison. $P$-values indicate the statistical difference ($P$-value ≤ 0.05 = *, $P$-value ≤ 0.01 = **, $P$-value ≤ 0.001 = ***, $P$-value ≤ 0.0001 = ****).

Similar to hypersaline marsh in Tristomo bay and hypersaline microbial mats in Guerrero Negro (65, 66), *Candidatus* Thermoplasmatota (class E2) MAGs were detected in the Solar Lake mats. This order was identified in Guaymas Basin hydrothermal vent (58). Asgardarchaeota superphylum MAGs were detected in all Solar Lake microbial mats and were previously reported as major players in the Shark Bay, Australia microbial mats (5, 67, 68). The detection of Asgardarchaeota MAGs carrying complete carbon fixation pathways and several glycoside hydrolases in several microbial mat systems (5, 67, 68) support the hypothesis that they play an important role in carbon cycling in anoxic mat systems (67, 69). Functional prediction of the Solar Lake Asgard MAGs suggests that they play a similar role (Table S6).

## Bacterial benthic community in the shallow versus exposed Solar Lake microbiome

*Candidatus* Omnitrophota MAGs were detected only in the shallow-water flat mats, whereas *Myxococcota* MAGs were specific to the exposed mats.

Members of *Candidatus* Omnitrophota are metabolically versatile and detected in diverse environments (70, 71). The majority of the *Candidatus* Omnitrophota members lack ROS response genes including the two MAGs assembled in our study (71). On the other hand, all *Myxococcota* MAGs had ROS response genes, explaining their presence in the exposed microbial community (Table S7). Unlike the sulfated exopolysaccharide (EPS) recycling *Myxococcota* members detected in the cyanobacterial mat ecosystems in Shark Bay (Australia) (72), the Solar Lake *Myxococcota* members possessed genes responsible for the degradation of cellulose, arabinose, and xylose (Table S7).

It is worth noting that the bacterial phylum *Chloroflexota* dominated the Solar Lake shallow-water and exposed mats similarly to the Shark Bay, Australian smooth mats (Fig. 1C; Fig. S2) (67). Phototrophic *Chloroflexota* MAGs (family: *Chloroflexaceae*) genetic makeup suggests that they play a role in anoxygenic phototrophy and carbon fixation using the Wood-Ljungdahl pathway. Since previous studies have shown that *Anaerolineae* are chemoorganotrophs and play a role in carbon degradation and mat stability (73, 74), *Anaerolineae* MAGs within *Chloroflexota* are likely to play a similar role in the Solar Lake.

## Phototrophy and carbon fixation in the Solar Lake exposed versus shallow-water microbiome

The taxonomic composition of *Cyanobacteria* within the shallow-water and exposed microbial mats exhibited similarities to the 1977 study by Krumbein and colleagues, where *C. chthonoplastes* and *Aphanothece halophytica* (current name: *Halothece* sp. PCC 7418) were detected in both studies (8). Nevertheless, Krumbein and colleagues showed that the summer shallow-water mats were predominantly occupied by coccoid *Cyanobacteria*, specifically *A. halophytica* (*Halothece* sp. PCC 7418) (8). However, our analysis detected only one MAG affiliated with *Halothece* (with a relative abundance of 0.46%) in the E-SO$_{shallow-water}$ sample (Table S4). Surprisingly, the filamentous *C. chthonoplastes* that were previously detected mainly in winter shallow-water mats (8) were detected in our summer shallow-water (N-SO and E-SO) and exposed (E-SO) mats with relative abundance ranging from 0.87% to 0.39%. Additionally, MAGs assigned to *C. chthonoplastes* had one of the highest PTR (peak-to-trough ratio) values in the studied Solar Lake benthic community, suggesting active replication in comparison to other community members (Fig. S5). Previous reports documented that *C. chthonoplaste* was able to resist $700 \mu E m^{-2} S^{-1}$ and produce preoxidase and superoxide dismutase (75), and several other genes were reported to be upregulated (76) to alleviate photooxidative stress. Such genes, glutathione peroxidase, carotenoid (*crtI* and *crtB*), and σ-factor/σE were detected in Solar Lake *C. chthonoplastes* MAGs (Fig. 6; Table S8). Our analysis challenges the previous notion, primarily based on visualization and cyanobacterial cell morphology, that filamentous *Cyanobacteria* are exclusively abundant in the Solar Lake winter microbial mats because of their sensitivity to photooxidative stress (8).

As expected, cyanobacterial MAGs possess genes for the Calvin cycle. Nonetheless, two *Chloroflexota* MAGs had the genetic potential to fix carbon using the Wood-Ljungdahl pathway. This is in contrast to most *Chloroflexota* that are known to use HP3 bi-cycle; the pathway by which *Chloroflexota* members usually fix carbon (77). The Solar Lake E-SO$_{exposed}$ aerobic anoxygenic photrophic MAG (AAP) *R. sodomensis* MAGs had a nearly complete 3-Hydroxypropionate bi-cycle gene set and the SOX complex operon, suggesting its autotrophic potential (78). Previous studies reported that *R. sodomensis* cannot grow autotrophically (79).

## Methanogenesis in the Solar Lake exposed and shallow-water flat mats communities

Methanogens and methanogenesis-related genes were detected in the four microbial mats samples. The absolute and relative abundance of *mcrA* followed similar trends. The number of *mcrA* copies was 2–3 times higher than methanogenic cell counts previously reported in the Solar Lake sediments (13). Most probably this is due to the presence of more than one *mcrA* copy in the genomes of the Solar Lake methanogens, and a limitation in the detection protocol of the previous study (13).

The identity of the Solar Lake benthic methanogens has not been previously explored (13). In this study, we defined methanogen MAGs as those MAGs that harbored ≥60% of the genes involved in methanogenesis based on DRAM annotation. We were unable to define methanogens based on the presence or absence of *mcrA* because this gene was poorly covered in our metagenomes and the metagenomes of similar environments (5). Putative methanogen MAGs were assigned to *M. evestigatum*, *Archaeoglobales*, *Candidatus* Methanofastidiosales orders, and an unclassified class within the phylum *Bathyarchaeia* (80–85). Similarly, the methanogenic community members *Methanosarcinales* and *Candidatus* Methanofastidiosales have been identified in the hypersaline microbial mats of Guerrero Negro desert region (66). Intriguingly, our study revealed that the previously reported Solar Lake 16S rRNA gene of Archaeal Cluster II (42) formed a cluster (bootstrap: 87%) with the 16S rRNA genes of *Archaeoglobales* MAGs in this study. This finding provides the first evidence that *Archaeoglobales* is one of the methanogenic lineages present in both the planktonic and shallow-water mat benthic communities of the Solar Lake (Fig. S6).

Giani and colleagues have shown that the major substrate used for methanogenesis by the Solar Lake benthic community is methylated amines, explaining their coexistence with sulfate reducers (13). Our data support this hypothesis based on (i) high relative abundance (CPM) of genes encoding methylotrophic methanogenesis via the utilization of methylated amines, with *mttb* as the specific biomarker, and (ii) all Solar Lake methanogen MAGs, except the MAG affiliated to *Candidatus Methanofastidiosales*, encoded the genes highly indicative of monomethylamine- or trimethylamine-based methanogenesis. In addition to methylotrophic methanogenesis, methanogens in the exposed and shallow-water flat mats have the genetic potential to perform either acetoclastic (*M. evestigatum* and *Candidatus* Methanofastidiosales) or hydrogenotrophic (*Archaeoglobales* and *Bathyarchaea*) methanogenesis. This is in agreement with the previous hypothesis that methylotrophic methanogenesis is the predominant methanogenic pathway but not the sole one, in hypersaline microbial mats (66, 86). Interestingly, besides methanogenesis, the methanogenic community in the Solar Lake had the genetic potential to reduce sulfate, oxidize thiosulfate, and fix nitrogen. The presence of *dsrA/B* genes in the Solar Lake *Archaeoglobales* MAGs might suggest single-cell sulfate-dependent anaerobic methane oxidation (AMO) as previously described with *Candidatus* Methanomixophus dualitatem (82–85, 87).

The Solar Lake exposed samples (N-SO$_{exposed}$ and E-SO$_{exposed}$) had lower moisture content (Table 1), and it is anticipated that more oxygen would penetrate the exposed sediments compared with the suboxic shallow-water sediments (18). Hence, it is likely that aerobic methane oxidation would be present in both exposed samples; however, we only detected it in N-SO$_{exposed}$. The inability to detect aerobic methanotroph (*pmoA*) in E-SO$_{exposed}$ is likely attributed to their low abundance, which often goes unnoticed in shotgun sequencing (5).

## Solar Lake-exposed and shallow-water mat communities' adaptation to summer exposure

The Solar Lake is characterized by a water salinity reaching 18% during the summer season (11). Hence, the Solar Lake benthic microbial communities are anticipated to possess salinity stress genes analogous to similar ecosystems (5, 88). Environmental changes such as water subsiding from microbial mats can profoundly affect microbial

community functional capacity due to oxygen intrusion, higher salinity, and drought. Few studies looked at the effect of tidal-driven desiccation or salinity gradients on the structure of similar cyanobacterial mats, but not on their function (21, 22).

Here, we showed that exposed flat microbial mats specifically N-SO$_{exposed}$ were enriched in dihydrogeodin oxidases/laccases (AA1), a Cazyme gene involved in the oxidation of recalcitrant organic matter such as polyphenolic. This is likely due to drought and oxygen intrusion, into this sample, as previously shown in other environments (24, 89). Moreover, N-SO$_{exposed}$ and E-SO$_{exposed}$ assemblies contained a higher number of ROS response genes. The low number of ROS detoxification genes in the flat shallow-water mats may be supported by the shallow-water suboxic conditions (DO = 2.7 mg/l) previously reported in the lake sediments (18). Moreover, the enrichment of genes involved in ROS response and the genetic potential for osmoprotectants production in the exposed mat N-SO could indicate that this mat is going through oxygen and drought stress (Fig. 6). To our knowledge, the effect of drought was not previously studied on hypersaline stromatolite-like systems. Nevertheless, the reported stress response employed by the Solar Lake microbial mat community, evident in the ROS and salinity response gene enrichment, is similar to other environments such as vegetated mudflat ecosystems (90), paddy soil (24), grassland rhizosphere (91), and alkaline wetland (92).

Our data show that there is an altered genetic makeup of the Solar Lake benthic microbial mat communities in the exposed versus shallow-water zones during the summer. Nevertheless, more sampling sites throughout the lake of exposed and shallow water are needed to ensure the robustness of our observations.

## Conclusion

In summary, our study sheds light on the benthic microbial communities thriving in the Solar Lake, Taba, Egypt. The microbial mats in this unique ecosystem exhibit remarkable structural and functional complexity. We stressed on the significant contribution of phototrophy and methanogenesis. Here, we unveiled key findings that collectively contribute to a deeper understanding of the Solar Lake microbial flat mat benthic community. These findings include an increase in ROS response genes and the unique identification of *Halobacteriota*, *Candidatus* Altiarchaeota, and *Myxococcota* in the exposed mats. This iterates the functional and structural adaptation to external environmental stressors such as oxygen intrusion and drought caused by the partial receding of water.

Moreover, we showed that the abundance and computed active replication of *C. chthonoplaste* indicate that this *Cyanobacteria* is not only a major player in the winter, as previously reported (8), but in the summer community as well. Additionally, methylotrophic methanogenesis is the most abundant methanogenic pathway in the Solar Lake mats, and the identified *Archaeoglobales* genetic makeup can contribute to AMO. Further studies are needed to understand the seasonal variation and microbial stratification in this unique Solar Lake ecosystem.

Given the current climate crisis, and the anticipated prolonged drought events in Sinai, Egypt, and the Middle East (93, 94), it is crucial to understand how partially exposed microbial mats respond to drought and oxygen intrusion. There is limited research on the effects of climate change on microbial mats ecosystems (95–97), and none has specifically addressed drought effects. Decoding the hypersaline microbial mats' genetic adaptation, particularly the exposed mats that are subjected to partial drought, might aid in developing adaptive management plans for maintaining or restoring such ecosystems.

## ACKNOWLEDGMENTS

This project was funded by the British Ecological Society "Ecologist in Africa" grant (Project number: EA19/1215).

Mr. Ayman Mohamed is acknowledged for his help during sample collection.

R.Z.A. and R.S. designed the study. R.Z.A., A.A., and M.M. collected the samples. R.Z.A. and S.F.A. carried out laboratory experiments. R.Z.A. and A.H.A.E. analyzed and interpreted the data. R.Z.A. performed statistical analyses and generated figures. R.Z.A. and R.S. wrote the manuscript. R.S. funded the project, except for the qPCR experiments that were undertaken in the W.L. laboratory. All authors discussed the results and commented on the manuscript at all stages.

## AUTHOR AFFILIATIONS

[1]Biology department, The American University in Cairo, Cairo, Egypt
[2]Department of Microbiology and Immunology, Faculty of Pharmacy, University of Sadat City, Sadat City, Egypt
[3]Microbiology and Immunology Department, Faculty of Pharmacy, Ahram Canadian University, 6th of October City, Giza, Egypt
[4]Max Planck Institute for Terrestrial Microbiology, Marburg, Germany

## AUTHOR ORCIDs

Rehab Z. Abdallah  http://orcid.org/0000-0001-7194-2861
Ali H. A. Elbehery  https://orcid.org/0000-0002-8028-9849
Werner Liesack  https://orcid.org/0000-0002-9533-1552
Rania Siam  http://orcid.org/0000-0002-2879-6368

## FUNDING

| Funder | Grant(s) | Author(s) |
| --- | --- | --- |
| British Ecological Society (BES) | EA19/1215 | Rania Siam |

## DATA AVAILABILITY

The metagenome shotgun sequencing raw data were deposited in the NCBI BioProject database under the accession numbers PRJNA1004913. Metagenome assemblies and metagenome assembled genomes (MAGs) used in this study were deposited in Zenodo repository at https://doi.org/10.5281/zenodo.10521873 (98). The data files names corresponding to each sample are indicated in Table S1.

## ADDITIONAL FILES

The following material is available online.

Supplemental Material

**Supplemental material (mSystems00095-24-S0001.docx).** Supplemental materials and methods and Figures S1 to S6.
**Table S1 (mSystems00095-24-S0002.xlsx).** Solar Lake shallow-water and exposed mat metagenome statistics.
**Table S2 (mSystems00095-24-S0003.xlsx).** Solar Lake shallow-water and exposed mat metagenome DRAM annotations.
**Table S3 (mSystems00095-24-S0004.xlsx).** Solar Lake shallow-water and exposed mat MAG summary.
**Table S4 (mSystems00095-24-S0005.xlsx).** The cyanobacterial MAG relative abundance in the Solar Lake shallow-water and exposed mats.
**Table S5 (mSystems00095-24-S0006.xlsx).** Nitrogen cycling genes distribution across the Solar Lake shallow-water and exposed archaeal and bacterial MAGs and the relative abundance of *nir* gene-containing MAGs.
**Table S6 (mSystems00095-24-S0007.xlsx).** Carbon fixation pathways, Cazyme, and short-chain fatty acids and alcohol conversion genes in Asgardarchaota.

**Table S7 (mSystems00095-24-S0008.xlsx).** Solar Lake flat mat *Myxococcota* MAGs annotation using DRAM.

**Table S8 (mSystems00095-24-S0009.xlsx).** Photooxidative stress gene in *Coleofasciculus chthonoplastes* MAGs.

## Open Peer Review

**PEER REVIEW HISTORY (review-history.pdf).** An accounting of the reviewer comments and feedback.

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
