## [Reviewer comments · mSystems]

Deciphering the Functional and Structural Complexity of the Solar Lake Flat Mat Microbial Benthic Communities.

Rehab Abdallah, Ali Elbehery, Shimaa Ahmed, Amged Ouf, Mohamed Malash, Werner Liesack, and Rania Siam

Corresponding Author(s): Rania Siam, American University in Cairo

Review Timeline:

Submission Date:	January 28, 2024
Editorial Decision:	March 14, 2024
Revision Received:	March 28, 2024
Accepted:	April 4, 2024

Editor: Ryan Newton

Reviewer(s): Disclosure of reviewer identity is with reference to reviewer comments included in decision letter(s). The following individuals involved in review of your submission have agreed to reveal their identity: Ana Camila Gonzalez-Nayeck (Reviewer #1)

Transaction Report:

DOI: <https://doi.org/10.1128/msystems.00095-24>

Re: mSystems00095-24 (Deciphering the Functional and Structural Complexity of the Solar Lake Flat Mat Microbial Benthic Communities.)

Dear Prof. Rania Siam:

Thank you for allowing us to review your team's work. Below you will find my comments, instructions from the mSystems editorial office, and the reviewer comments.

Thank you for your detailed responses to the review comments. The reviewers believe these responses generally addressed their previous concerns. Both reviewers added a few additional comments, and these are indicated in their reviews. Please address these new reviewer comments/suggestions in your follow-up response and revised manuscript.

Also note on L585 - I believe there is a repeated sentence that should be removed.

Revision Guidelines

Sincerely,
Ryan Newton
Editor
mSystems

Reviewer #1 (Comments for the Author):

This study provides a genomic characterization of microbial Mats from Solar Lake, Taba, Egypt, which is a microbial mat system that is often cited in the literature but has been primarily characterized previously using conventional microbiology techniques. The submission provides a clear discussion of the significance and importance of both genomic data and the overall study. Significant findings highlight unique adaptations (e.g., Cyanobacteria equipped to handle photooxidative stress) to drought, which are likely to become more relevant as climate change continues. Below are non-exhaustive examples of two more places where the manuscript does a good job of clarifying the motivations and significance of the work:

Lines 115-117, 134-146, 138-140: These additions significantly clarify the motivations of the study. Great work.

Lines 606-610: Wow! This is a very high quantity of light for Cyanobacteria (most lab-grown cyanos suffer above $\sim 120 \mu\text{E m}^{-2} \text{S}^{-1}$), but this makes sense given your unique environment and this discussion point does a good job of underscoring the adaptations necessary to survive in such an environment. I agree with lines 612-616 that your analysis challenges the previous notion; I would have guessed this was simply too much light for Cyanobacteria.

Minor edits:

General: Cyanobacteria is always capitalized because it is a phylum. However, cyanobacterial (the adjective) should stay lowercase.

Line 527: incorrect use of "iterating," instead try "emphasizing" or "highlighting".

Line 575-577: if you are trying to say that Chloroflexota dominated in Solar Lake and they also dominated in Shark Bay, change "similar" (line 576) to either "similarly" or "as in."

If you are trying to say that the physical characteristics of the shallow-water and exposed mats in solar lake are similar to the physical characteristics of the smooth mats in Shark Bay, keep your sentence as is.

Line 583: "chemoorganotrophs" instead of "chemoragnotrophs"

Lines 742-751: Potentially move to the end? This paragraph is a more general ending than the next paragraph.

Reviewer #2 (Comments for the Author):

The reviewer appreciates the thoughtful revisions, especially the hypotheses and connections to the existing literature, that authors made to their manuscript.

I have just a few clarifications I would like to see:

1. In lines 153-154, the East samples are noted as being covered with mountain shade. Are these sites always shaded or does it see some direct exposure to the sun during the day?

2. Line 288. Should be Deseq2

3. Lines 526-529. While the potential HGT is interesting, I think it's also worth noting if these 2 candidate phylum have also been observed in non-aquatic systems? Have they been observed in other microbial mats?

Dear Professor Dr. Ryan Newton,

Thank you for reviewing our revised manuscript. We would also like to thank both reviewers and yourself for considering and revising our manuscript, allowing the careful analysis and representation of our work.

We have revised the manuscript according to the comments and requested modifications. Our modifications are clarified in this point-by-point response letter and highlighted in the text in green.

Again thank you for your time and consideration. We are looking forward to the published manuscript in mSystems.

Kind regards-Rania Siam

Subject: mSystems00095-24 Decision Letter

Re: mSystems00095-24 (Deciphering the Functional and Structural Complexity of the Solar Lake Flat Mat Microbial Benthic Communities.)

Dear Prof. Rania Siam:

Thank you for allowing us to review your team's work. Below you will find my comments, instructions from the mSystems editorial office, and the reviewer comments.

Thank you for your detailed responses to the review comments. The reviewers believe these responses generally addressed their previous concerns. Both reviewers added a few additional comments, and these are indicated in their reviews. Please address these new reviewer comments/suggestions in your follow-up response and revised manuscript.

Also note on L585 - I believe there is a repeated sentence that should be removed.

Thank you for your comment. We have removed the repeated sentence (line 583).

Revision Guidelines

Publication Fees: For information on publication fees and which article types are subject to charges, visit our <https://journals.asm.org/publication-fees> target="blank">website. If your manuscript is accepted for publication and any fees apply, you will be contacted separately about payment during the production process; please follow the instructions in that e-mail. Arrangements for payment must be made before your article is published.

Sincerely,
Ryan Newton
Editor
mSystems

Reviewer #1 (Comments for the Author):

This study provides a genomic characterization of microbial Mats from Solar Lake, Taba, Egypt, which is a microbial mat system that is often cited in the literature but has been primarily characterized previously using conventional microbiology techniques. The submission provides a clear discussion of the significance and importance of both genomic data and the overall study. Significant findings highlight unique adaptations (e.g., Cyanobacteria equipped to handle photooxidative stress) to drought, which are likely to become more relevant as climate change continues. Below are non-exhaustive examples of two more places where the manuscript does a good job of clarifying the motivations and significance of the work:

Lines 115-117, 134-146, 138-140: These additions significantly clarify the motivations of the study. Great work.

Thank you so much for your comments. We are glad that the modifications have clarified the motivation and goal of the study.

Lines 606-610: Wow! This is a very high quantity of light for Cyanobacteria (most lab-grown cyanos suffer above $\sim 120 \mu\text{E m}^{-2} \text{S}^{-1}$), but this makes sense given your unique environment and this discussion point does a good job of underscoring the adaptations necessary to survive in such an environment. I agree with lines 612-616 that your analysis challenges the previous notion; I would have guessed this was simply too much light for Cyanobacteria.

Thank you so much for your comment.

Minor edits:

General: Cyanobacteria is always capitalized because it is a phylum. However, cyanobacterial (the adjective) should stay lowercase.

Thank you, we have made the required modifications.

Line 527: incorrect use of "iterating," instead try "emphasizing" or "highlighting".

Thank you for your comment. We have changed "iterating" to "emphasizing" (line 530).

Line 575-577: if you are trying to say that Chloroflexota dominated in Solar Lake and they also dominated in Shark Bay, change "similar" (line 576) to either "similarly" or "as in."

If you are trying to say that the physical characteristics of the shallow-water and exposed mats in solar lake are similar to the physical characteristics of the smooth mats in Shark Bay, keep your sentence as is.

Thank you for your comment. We have changed "similar" to "similarly" (line 582).

Line 583: "chemoorganotrophs" instead of "chemoragnotrophs"

We made the suggested edit (line 587).

Lines 742-751: Potentially move to the end? This paragraph is a more general ending than the next paragraph.

We have moved the paragraph to lines 765-774 .

Reviewer #2 (Comments for the Author):

The reviewer appreciates the thoughtful revisions, especially the hypotheses and connections to the existing literature, that authors made to their manuscript.

I have just a few clarifications I would like to see:

1. In lines 153-154, the East samples are noted as being covered with mountain shade. Are these sites always shaded or does it see some direct exposure to the sun during the day?

Thank you for your comments. During the sample collection time (12:00-13:00) on 10/7/2021, the sampling location was covered by mountain shades. Nevertheless, we believe that during the Sun's movement to the west (for sunset), these spots will be exposed to sunlight. We have modified the text and included details for clarification (line 155).

2. Line 288. Should be Deseq2

We have made the required modifications (line 290).

3. Lines 526-529. While the potential HGT is interesting, I think it's also worth noting if these 2 candidate phylum have also been observed in non-aquatic systems? Have they been observed in other microbial mats?

Both *Ca. Aenigmarchaeota* and *Bathyarchaeota* were detected in other microbial mats, deep-sea, and coastal sediments, including freshwater and groundwater sediments. Up to our knowledge, both *Ca. Aenigmarchaeota* and *Bathyarchaeota* were not previously documented in soil ecosystems. It is worth noting that on the 16S rRNA gene level, *Ca. Aenigmarchaeota* was detected in limited percentages in soil ecosystems (Li, 2021-Reference 56 in the manuscript). We have modified the manuscript text to reflect the additional information (lines 531-534).

Re: mSystems00095-24R1 (Deciphering the Functional and Structural Complexity of the Solar Lake Flat Mat Microbial Benthic Communities.)

Dear Prof. Rania Siam:

Your manuscript has been accepted, and I am forwarding it to the ASM production staff for publication. Your paper will first be checked to make sure all elements meet the technical requirements. ASM staff will contact you if anything needs to be revised before copyediting and production can begin. Otherwise, you will be notified when your proofs are ready to be viewed.

Cover Image Submissions: If you would like to submit a potential Cover Image, please email a file and a short legend to msystems@asmusa.org. Please note that we can only consider images that (i) the authors created or own and (ii) have not been previously published. By submitting, you agree that the image can be used under the same terms as the published article. Image File requirements: TIF/EPS, 7.5 inches wide by 8.25 inches tall (at least 2,250 pixels wide by 2,475 pixels tall), minimum 300 dpi resolution (600 dpi preferred), RGB, and no figure elements, e.g., arrows or panel labels. The legend should be a short description of the image, 1-2 sentences recommended.

Sincerely,
Ryan Newton